# Numerical Flow Simulation on the Virus Spread of SARS-CoV-2 Due to Airborne Transmission in a Classroom

**DOI:** 10.3390/ijerph19106279

**Published:** 2022-05-22

**Authors:** Lara Moeller, Florian Wallburg, Felix Kaule, Stephan Schoenfelder

**Affiliations:** Faculty of Engineering, Leipzig University of Applied Sciences, 04277 Leipzig, Germany; lara.moeller@stud.htwk-leipzig.de (L.M.); florian.wallburg@htwk-leipzig.de (F.W.); felix.kaule@htwk-leipzig.de (F.K.)

**Keywords:** SARS-CoV-2, indoor air, ventilation, CFD, infection risk, airbone transmission, public health, environmental science and engineering

## Abstract

In order to continue using highly frequented rooms such as classrooms, seminar rooms, offices, etc., any SARS-CoV-2 virus concentration that may be present must be kept low or reduced through suitable ventilation measures. In this work, computational fluid dynamics (CFD) is used to develop a virtual simulation model for calculating and analysing the viral load due to airborne transmission in indoor environments aiming to provide a temporally and spatially-resolved risk assessment with explicit relation to the infectivity of SARS-CoV-2. In this work, the first results of the model and method are presented. In particular, the work focuses on a critical area of the education infrastructure that has suffered severely from the pandemic: classrooms. In two representative classroom scenarios (teaching and examination), the duration of stay for low risk of infection is investigated at different positions in the rooms for the case that one infectious person is present. The results qualitatively agree well with a documented outbreak in an elementary school but also show, in comparisons with other published data, how sensitive the assessment of the infection risk is to the amount of virus emitted on the individual amount of virus required for infection, as well as on the supply air volume. In this regard, the developed simulation model can be used as a useful virtual assessment for a detailed seat-related overview of the risk of infection, which is a significant advantage over established analytical models.

## 1. Introduction

In order to keep the SARS-CoV-2 pandemic at bay, the restriction of possible transmission routes is and remains the decisive method, as it is not yet understood how effectively acquired immunity will actually limit further spread—especially with regard to mutants. Thus, SARS-CoV-2 could continue to influence social life for many years to come. Possibly for decades, according to the World Health Organization [1].

While airborne transmission was still critically discussed in the early phase of the pandemic [2] (and before concerning SARS [3]), it is now considered certain that, in contrast to transmission via contaminated surfaces [4], airborne transmission is the primary transmission mechanism of SARS-CoV-2 [5,6,7,8,9,10,11,12,13]. At the same time, the avoidance of this type of transmission—compared to smear and droplet infection—is difficult to realise in terms of the necessary restrictions on public life. Large droplets sink to the ground within a radius of r=1.5m around infectious persons (emitted by talking, sneezing, coughing, or general breathing) [14]. Aerosols spread throughout the room and can still be infectious for more than 3h after aerosolisation [15]. In rooms where people spend several hours in one place, such as computer rooms or libraries, barriers are used to reduce the risk of infection. Anyway, those barriers do not prevent airborne transmission of the SARS-CoV-2-virus-laden aerosol particles [16]. The virus concentration in the occupied area must be decreased by appropriate measures to ensure a low risk of virus transmission [17,18].

Intrinsically, there are great concerns about the use of indoor rooms as this drastically increases the critical radius around an infectious person [15], which is why a permanent exchange of air with unpolluted outdoor air is mandatory [19]. Concerning the winter months, this leads to a conflict between sufficient air exchange with the outside air and the additional economic and ecological aspects with regard to heating energy in order to maintain a comfortable indoor climate. This situation is particularly critical for heavily frequented indoor spaces, whose availability is given high priority even during pandemic situations: for example, schools for maintaining face-to-face teaching. Therefore, in addition to fundamental research to tackle the SARS-CoV-2-virus itself, the prevention of infections, in general, is also an important topic of current research.

The first models describing SARS-CoV-2 viral contamination of indoor spaces have been based on analytical calculation approaches. Depending on the model, the calculations are refined with various boundary conditions, such as the number of infectious persons, the influence of masks, information on the duration of stay in the room, the viral lifespan, or with general ventilation conditions [20,21]. The model of Hartmann and Kriegel [21] estimates the indoor airborne infection transmission from carbon dioxide concentration, which is a very practicable way, as Rudnick and Milton [22] already stated. Using these models, it is possible to estimate the risk of infection for a variety of scenarios, e.g., for visits to restaurants or theatres, training at the gym or working in open space offices [23]. Although these models provide a good initial overview, they cannot be used to spatially resolve virus concentrations in a room. Especially in ventilated rooms, this is to be considered critical as contaminated air can be removed very efficiently through a well-conceived fluid flow design. However, numerical methods such as computational fluid dynamics (CFD) can be used for such cases. Even though scientific contributions concerning the airborne transmission of SARS-CoV-2 based on the computational fluid dynamics method in classrooms exist [24,25,26,27,28]; however, there is actually no work that aims to provide a temporally and spatially-resolved risk assessment with explicit relation to the infectivity of SARS-CoV-2 in classroom environments. The only known work by Shao et al. [29] gives an overview of the potential risk of infection based on the findings of Wan et al. [30] and Morawska et al. [31] in different scenarios (elevator, supermarket, classroom), but does not use SARS-CoV-2-specific data for infection risk assessment.

In this work, a model is proposed based on computational fluid dynamics considering virus particle concentration, which is able to quantify the infection risk.

## 2. Model Approach

### 2.1. Modelling of the Viral Load

In the following section, the modelling approach for calculating the indoor SARS-CoV-2 virus spread in the presence of one infectious person is presented. The contaminated indoor air basically consists of two gas phases: one gas phase has the properties of air, and the other gas phase is defined by a so-called tracer gas. The distribution of these two gas phases is not calculated by an analytically homogenised model, but rather using the method of computational fluid dynamics (CFD). In this method, the five conservation equations of fluid mechanics for mass (one eq.), momentum (three eqs.), and energy (one eq.), the so-called Navier–Stokes equations, are approximated by numerical methods. For this purpose, the model space is discretised into a large number of control volumes (so-called meshing). In each control volume, the conservation equations are numerically solved, which allows a very good temporal and spatial resolution of the results. Commercial CFD software (more precisely ANSYS Fluent) is used in the context of this work. At this point, reference is made to fundamental works on the method [32,33]. In computational fluid dynamics, the dilution of any gas phase can be measured, so the choice of tracer gas is arbitrary. For the conducted numerical simulations, a tracer gas with properties similar to those of water vapour is chosen. A resulting concentration of this gas phase in a control volume contains a virus particle count in the evaluation depending on the infectious person’s activity level.

The virus particle count in each control volume can be calculated from the ratio of the mass concentration of contaminated air emitted by the infectious person to the representative viral load according to activity level. In previous published studies, the viral load of an infectious person was investigated or assumed at different levels of activity, such as breathing and speaking. The measurements of Hartmann et al. [34] show an emission of 134 aerosol particles/s when breathing and 195 aerosol particles/s when speaking. Assuming that the typical diameter of a fine aeorosol particle leaving the mouth during breathing and speaking is about 5 µm [20], a volume of $5.2\times 10^{-10}\;\mathrm{mL}$ is obtained for a droplet. With 134 aerosol particles/s, this results in a volume flow of $7.01\times 10^{-8}\;\mathrm{mL}/s$ during breathing. However, viral load density varies widely and can range from $10^{4}$ to $10^{11}$ (extremely infectious) SARS-CoV-2 virus particles/mL [35,36,37,38,39] (It should be noted at this point that the term ‘SARS-CoV-2-virus particle/s’ used in this article is identical to ‘SARS-CoV-2 RNA copy/s’). Thus, a particle emission range of almost 0 to about 7000 SARS-CoV-2-virus particles/s follows. Therefore, during speaking, emissivity rates of even more than 10,000 SARS-CoV-2-virus particles/s can consequently be achieved. However, Lelieveld et al. [20] evaluated several studies and came to the conclusion that the viral density concerning the original variant (wildtype) is often in the range between $10^{8}$ and $10^{9}$ SARS-CoV-2-virus particles/mL. This results in values of around 100 SARS-CoV-2-virus particles/s in case of speaking, which is also in good agreement with assumptions of Kriegel et al. [40], who quantified an output of infectious units with 100 particles/s. (The work does not distinguish between different virus variants. It is therefore assumed that any assumption refers to the original virus type (wildtype)).

The quantity of emitted SARS-CoV-2-virus particles/s (RNA copies/s) could also be determined by measurements of Coleman et al. [41]: approx. 10.8 SARS-CoV-2-virus particles/s are emitted when breathing and 81.6 SARS-CoV-2-virus particles/s when speaking, on average. (In Coleman et al. [41], the emitted viral load is given per breath. The values given here are per second, assuming the duration of one breath to be 5 seconds and noting that fine aerosols account only for 85.4% of the detected viral load as mentioned by the authors). The study by Coleman et al. [41] included patients infected with different viral variants. Newer virus variants may have higher emission rates than older variants, but due to the low sample size it was not possible to determine aerosol-shedding patterns depending on the type of variant. In any case, these values are within the range of Lelieveld’s assumptions of 100 SARS-CoV-2-virus particles/s during speaking and 10 during breathing concerning the original variant (wildtype), respectively. Following the studies presented, within the work presented here, a viral emission of 10 SARS-CoV-2 virus particles/s during breathing and 100 SARS-CoV-2 virus particles/s for speaking is assumed.

This viral load of an infectious person is adapted to the average respiration rate of a person with $0.36\;m^{3}/h$ [42,43]. An average respiratory volume of 0.5L/breath and 12 breaths/min [44] was assumed. Under the assumption of a breathing in an open area of $2\times 10^{-4}\;m^{2}$, this results in a flow velocity of 0.5 m/s. Thermal effects, such as rising and sinking of the gas, are taken into account. For the calculation of this multispecies flow, the species transport model implemented in ANSYS Fluent is used.

The concentration of tracer gas in a breath of an infectious person is chosen in the range of hundreds of parts per million. It is scaled as a multiple of the real mass concentration ($10^{-4}\;{\mathrm{kg}}_{\mathrm{viral}\;\mathrm{load}}/{\mathrm{kg}}_{\mathrm{air}}$ instead of $10^{-10}$ for speaking and $10^{-11}\;{\mathrm{kg}}_{\mathrm{virus}\;\mathrm{particle}}/{\mathrm{kg}}_{\mathrm{air}}$ for breathing) of aerosol particles in order to keep numerical errors as low as possible. Concentration differences between the contaminated indoor air and fresh air (no virus mass concentration) change exponentially over time. Variations at different species mass concentrations in infectious respiration from $10^{-4}\;{\mathrm{kg}}_{\mathrm{viral}\;\mathrm{load}}/{\mathrm{kg}}_{\mathrm{air}}$ to $10^{-2}\;{\mathrm{kg}}_{\mathrm{viral}\;\mathrm{load}}/{\mathrm{kg}}_{\mathrm{air}}$, such as CO_2_ in Hartmann and Kriegel [21], results in deviations between 2–3%.

### 2.2. Assessment of Infection Risk

The simulation results can be used to determine the number of SARS-CoV-2-virus particles that are inhaled in total (resulting virus load) over time at fixed positions. With the resulting virus load (sum of inhaled particles) for that positions, the risk of infection for each seat can be assessed at any time.

SARS-CoV-2-virus particles are counted in time intervals of 10 s, whereas the concentration of the species is evaluated in a volume of size $1\times 10^{-3}\;m^{3}$ at each position of the susceptible persons at the level of the inhalation area. This volume has the shape of a sphere in front of the mouth and it is assumed to be completely inhaled. The particles are summed up over time as a floating point number for each inhalation volume. Wearing a mask over the mouth and nose with filtering effect is not taken into account. Particles counted in a control volume are not removed from the computational grid afterwards in order to avoid further scripting slowing down the numerical solver. The effects of this treatment is estimated to be small: for a room volume of $240\;m^{3}$ ($80\;m^{2}$ and 3m room height), the ratio of an inhalation volume to the room volume is 1:240,000. This means that there are significantly more particles in the room than in the inhalation volume that should be removed. Convection and diffusion processes would generally ensure that there are no large concentration gaps in the room. In most cases, however, the particles are not inhaled twice due to the large number of particles and the volume of the room.

Together with the D50 value, the individual risk assessment can now be made. Thereby, the D50 value indicates the mean dose of inhaled viral copies that cause infection in 50% of non-vaccinated individuals. However, this parameter is not clearly known for SARS-CoV-2, so it must be assumed. Nevertheless, the correct range of values is not important for modelling the airflow, as the evaluation of infection risk is conducted after the solution calculation (post-processing). This means that the threshold for high risk of infection (D50) can be easily adjusted as new research data become available. Based on [20], the D50 value is assumed to be in the range of 100–1000 viral copies. For the evaluation of the present results, a cumulative inhaled particle count of 500 particles is considered as a high risk for infection with the wildtype variant of SARS-CoV-2 as suggested by Riediker and Monn [45]. Newer virus variants such as Delta and Omikron have been found to have increased transmissibility properties [46,47]: Delta and Omikron, for example, show twofold and fourfold higher transmissibility, respectively, than the wild type [46].

The increase in transmissibility from the wildtype (used virus variant in the presented study) to other virus variants could be incorporated into numerical modelling by lowering the critical virus load (e.g., the critical dose for the delta type may be assumed to be 300 particles, for omicron even only 100 particles could be denoted as a critical viral load [48]).

## 3. Application Model

### 3.1. General Description

The approach presented in this paper for a spatially and temporally resolved description of viral load is applied to a classroom typically found in universities or schools. Figure 1a shows the model of the ventilated room. The window area is located in the upper part of the room opposite the door, which is located on the other side of the room. Supply air enters the room through a $1\;m^{2}$ window area. Indoor air can only exit the room through the door opening, which is an outlet area of $0.15\;m^{2}$.

It is assumed that there are 16 persons (one infectious lecturer and 15 susceptible persons) in a room with a floor area of $80\;m^{2}$(10m×8m) and a ceiling height of 3m. Each person has a radius of at least r=1.5m around themself, in which no other person is present. Consequently, it is considered that general distance rules are followed; thus, the spread mechanism is based exclusively on airborne transmission. According to Figure 1b, the rows are numbered from 1 to 5 in the x-direction during evaluation, whereas row 1 is closest to the lecturer. In the z-direction, the position of the seats is described as being close to the window (w), close to the door (d), or referred to as the middle area in between, denoted as centre (c).

At time t=0, there is only virus-free air in the room in each configuration examined. Furthermore, the temperature distribution is homogeneous, no one is breathing or speaking, and the windows are closed. In the first time step, the virus source then begins to emit contaminated air at a temperature of 310.15 K (37 °C), and the windows are opened. The incoming air has a temperature of T = 298.15 K (25 °C); this setting represents ventilation in summer. The surface temperature of the persons is 310.15 K (37 °C). The room is surrounded by six stationary walls, which are thermally adiabatic to the outside.

The model is meshed with approx. 2.6 million elements and is refined at near-wall regions in order to obtain acceptable $y^{+}$ values. For the near-wall treatment of the room walls that are parallel to the flow direction (from the window to the door) and for those of the bodies of people in the room, the first near-wall node is set to $y^{+}\approx 1$ (the mesh is fine enough to resolve the viscous sublayer). A transient modelling approach is used, which determines the viral load every 0.5s over a period of t=90min.

In the analysed situation, the lecturer is infectious and emits virus particles. Both the case of an active teaching situation (infectious lecturer is speaking) and the case of an examination situation (infectious lecturer is breathing) are simulated. These cases are considered for a sparsely ventilated and a highly ventilated room. Note that the supply air volume flow in the case of high ventilation ($432\;m^{3}/h$) is within the suggested supply air volume flow of 25–30 $m^{3}/\left(h\cdot \mathrm{person}\right)$ for school buildings [49], which is clearly above the minimum ventilation rate for classrooms of $18\;m^{3}/\left(h\cdot \mathrm{person}\right)$ given in ANSI/ASHRAE [50]. Additionally, this value is above the recommended outdoor airflow of about $25.2\;m^{3}/\left(h\cdot \mathrm{person}\right)$ to maintain ${\mathrm{CO}}_{2}$-levels less than 1000 ppm [51]. The analysed ventilation and viral loading configurations are given in Table 1.

### 3.2. Numerical Treatment of Air Flow

No fixed flow pattern (laminar flow) is to be expected due to the supply of airflow via the window, the inlet via the breathing of the infected person, and the buoyancy-induced flow [52] due to the temperature difference between persons and room air. Since a resolution of all the friction lengths of the vortices would result in too much computational effort, Reynolds-averaged Navier–Stokes equations are used. For flow modelling, the common RANS-based *k*-ϵ model is used (averaging operation to the Navier–Stokes equations (mass, momentum, energy) to obtain the mean equations of fluid flows called Reynolds Averaged Navier–Stokes (RANS) equations). However, the transport equations for the turbulence quantities *k* and ϵ are not treated as in the standard model; instead, they are developed with the help of the so-called renormalisation group theory (RNG), which offers an analytical treatment of the effective viscosity. Since flow regions with lower Reynolds numbers can be better taken into account from this, the model is additionally supported by an enhanced wall treatment, which represents the viscous sublayer more precisely, e.g., boundary regions, as in the present case in the area of the head, can thus be better resolved and the viral concentrations can generally be determined in more detail. All parameters and their values used in our model are summarised in Table 2. For a more extensive fluid mechanical description of the *k*-ϵ (RNG) model, please refer to [53].

## 4. Results

In the following, a comparison of the results obtained for the analysed classroom scenarios after a 90 min lesson is given. Figure 2 shows the resulting virus loads for each seat in the room under low ventilation (Figure 2a,c) and high ventilation conditions (Figure 2b,d) if the infectious lecturer is breathing only (Figure 2a,b) or speaking (Figure 2c,d). This lecturer is assumed to emit 10 and 100 SARS-CoV-2-virus particles per second, respectively, as mentioned in Section 2.1; the unit is from now on abbreviated as parts/s to denote emitted viral load. Figure 2a,b indicate lower resulting virus loads of susceptible students in the room compared to Figure 2c,d. The lower viral load of the infectious lecturer leads to a lower inhaled virus particle count at each seat in the room under the same ventilation conditions. The critical limit to a high infection risk of a total of 500 inhaled virus particles is not reached at any seat after t=90min if the infectious lecturer is breathing. The inhaled particle count remains below 100 for high and low ventilation at each seat if the infectious lecturer is breathing. It can be seen that at low ventilation, the C2, W2, and W3 positions have a higher resulting virus load after t=90min than the other positions. The high concentration of SARS-CoV-2 particles in the front rows of the classroom in the case of low ventilation can also be seen qualitatively in Figure 3.

Figure 2c,d show that higher ventilation (d) after t=90min ensures a lower resulting virus load in the room compared to lower ventilation conditions (c). A smaller number of inhaled particles can be detected at each position. At positions C1 and W1, this is not visually obvious, but even here, the values are dropping from (430 to 320 (C1) and 480 to 320 (W1)).

In the case of high ventilation, the resulting viral load is highest at positions C1 and W1 and lowest in the two back rows. At low ventilation, the back row is also the least contaminated with the virus. However, the highest virus loads after t=90min occur in the second row at seats C2 and W2, as well as W3. These three seats are the only ones where the critical limit of 500 inhaled particles after t=90min at low ventilation is reached. According to the determination in Section 2.2, the persons sitting there are assessed as having a high risk of infection.

Figure 4 shows the total number of particles inhaled over time at each seat if the infectious lecturer is breathing. The results at high and low ventilation are plotted on a graph for each seat to compare. It can be seen that the reduction in particle count due to high ventilation varies after t=90min at each seat: The relative change is largest in the back rows (between 67% and 75%). In the first row, the reduction of the resulting virus load is lowest (between 22% and 33%). In quantitative terms, the reduction of the resulting virus load is lowest in the first row. In contrast, the highest absolute reduction is seen at positions W2 and W3. In the case of high ventilation, the resulting virus load is highest in the first row and decreases with distance from the source of infection. At low ventilation, positions C2 and C3 have a higher viral load than C1. Furthermore, positions W2 and W3 have a higher viral load than W1. In both ventilation cases, the door side in the first three rows is less loaded than the window side or the centre. Interestingly, in contrast to low ventilation, high ventilation in the classroom leads to an increasing resulting virus load in the early time steps at some seats.

A lower resulting virus load at high ventilation compared to low ventilation occurs at these seats from different points in time: In the first row at C1, the resulting virus load drops below that of the low ventilation only after more than t=1h. A reduced total particle count due to the higher ventilation is determined at position D1 after almost t=1h, D2 after approx. t=50min, C2 and D3 after approx. t=40min, and W1 after approx. t=30min. At all other seats, this effect occurs after less than t=30min or is not detectable at all.

Figure 5 shows the resulting virus load as a function of time at low and high ventilation at different positions in the classroom if the infectious lecturer is speaking. The same described effects can be observed as in Figure 4. In contrast to the previously studied case with an infectious lecturer breathing, the critical limit concerning a high risk of infection is now reached for low ventilation conditions if the infectious lecturer is breathing—see Figure 2c. Figure 5 shows that at positions C2 and W3, a total of 500 particles are inhaled after just t=90min. At position W2, there is already a high risk of infection about 10 min earlier. It can also be seen that at seat C3, the limit value of 500 inhaled particles is just marginally not reached after t=90min. The resulting virus load is similar to seat C2. The result at W1 is also close to the limit of a high risk of infection after t=90min.

## 5. Discussion

### 5.1. Effect of Supply Air Volume on the Risk of Infection

In this work, the effects of different supply air volumes as well as of different viral load (breathing, speaking) of an infectious lecturer on the risk of infection at each seat for a classroom with 16 non-vaccinated persons is analysed. Within the considered period of t=90min, the limit of 500 inhaled particles regarding a high risk of infection is only reached in the case of a speaking infectious lecturer and low ventilation. The results indicate that high ventilation has significant advantages compared to low ventilation. For the selected boundary conditions (emitted viral load, D50 value), it can be concluded that the increase in supply air volume always means a longer low infection risk for the susceptible persons in the classroom. However, high ventilation does not immediately lead to a reduction of the resulting virus load at every seat. At some seats, the total inhaled particle count is slightly significantly reduced, not reduced at all, or even increased up to a certain time (up to more than t=1h, see Figure 4 and Figure 5), although the window side in each row is basically affected slightly less. In general, the effect of high ventilation on the resulting viral load appears to tend to increase with increasing distance from the virus source. A higher resulting virus load at the beginning with high ventilation than with low ventilation can be explained by a rapid distribution and mixing of the indoor air. If there is no or very low ventilation, the indoor air is distributed more inertly, including the virus-laden air emitted by the infectious lecturer. In the cases considered and the conditions specified, this does not result in a high risk of infection being reached sooner at any seat by using a high rather than a low ventilation. However, this may change with varying boundary conditions, e.g., more viral or high-emitting sources (so-called super-spreading) and/or a low personally required infectious dose. Although this difference does not seem particularly severe at low viral loads, a rapid distribution of viral particles at high ventilation could be problematic if the viral emission is very high as the infectious dose is reached quickly for a large number of people. In the case presented, after t=65min at the latest, the dilution effect of more fresh air per time in the room ensures that the virus load is lowered.

After t=90min, several conclusions can be made from the resulting virus load at each seat (see Figure 4 and Figure 5) at different ventilation levels: The relative reduction in the number of particles after t=90min is lowest in the first row, followed by the second row compared to the other rows. The largest absolute impact of high ventilation in each row can be detected at the window seat. The order of seats from highest to lowest resulting virus load changes with respect to high and low ventilation. It can be assumed that the type and direction of indoor airflow, such as cross-ventilation, displacement ventilation, mixed ventilation, etc., has a major influence on the temporal and spatial spread of SARS-CoV-2.

This shows very clearly why an investigation of the available and utilised ventilation concept for the spatially resolved infection risk is quite reasonable. Using the model presented, the reduction in inhaled particles and impact on the risk of infection can be quantified for each position in a room. This becomes particularly interesting when air cleaning equipment is used. Here, the optimal effect of the device(s) in the room can be determined in advance with a specific computational fluid dynamics design. Furthermore, the presented approach can be used to calculate the risk of infection for other types of indoor airflow situations as well as completely different rooms. The viral load of the infectious lecturer and the threshold for high risk of infection assumed in the paper are based on current scientific knowledge. However, as new scientific knowledge is gained, these values can be easily adjusted in the model. Likewise, the location of the infectious lecturer in the numerical model can be modified to solve specific problems.

### 5.2. ‘Green Classroom’ Situation

For evaluation purposes regarding the explained findings, the risk of infection is calculated for the case that the same room is located outdoors. All other conditions being equal, the room is no longer surrounded by walls on two sides, but by an inlet and outlet. For a common wind speed in Leipzig (Germany) of 4 m/s, a volume flow rate more than 1000 times higher than for the case with high ventilation is reached. The supply air will flow into the room over the entire wall surface. The calculated particle count in the inhalation volumes is in the range of $1\times 10^{-10}$ at each time step evaluated, so the cumulative particle count for this case is after t=30min, well below 1 at each position. This means that there is no high risk of an aerosol-based infection for the “outdoor classroom” at any time.

### 5.3. Comparison with Available Models

Based on the assumptions in Lelieveld et al. [20], the risk of infection can be calculated in the risk calculator of the Max Planck Institute for Chemistry (MPIC) [54] for different classroom situations. Note that the comparisons discussed below are also summarised in Table 3 and Figure 6. In case of a speaking infectious person and an air change rate of $0.146\;h^{-1}$, which corresponds to the low ventilation in the model presented here, there is a probability of 86% that at least one non-vaccinated person will be infected after t=1.5h. Concerning a high air change rate of $1.800\;h^{-1}$, the probability decreases to 46%. For a breathing infectious person, the probability is 9.4% for low ventilation and 3.0% for high ventilation. A probability of ≥50% such as 86% for one non-vaccinated person becoming infected in the case of low ventilation and emitted viral load through speaking (100parts/s) is understood as an infected person. Consequently, a probability ≤50% is equal to a non-infected state. It should be kept in mind that the results of the used risk calculator implies that a highly infectious person has been in the room for some time before the others: More than t=4h for low ventilation and more than t=1h for high ventilation.

From the numerically obtained results of the work presented here, it can be deduced that in the case of low ventilation and a breathing infectious lecturer emitting 10parts/s, a cumulative particle count of 500 is not exceeded at any position in the room after t=1.5h. For the higher viral load of 100parts/s due to speaking, the limit of the cumulative inhaled particle count of 500 virus particles is reached within t=1.5h at 3 seats. Within the investigated period of t=1.5h, no high risk of infection is found for both particle emission rates during high ventilation. However, it must be taken into account that in the numerical model presented here, the classroom is initially virus-free. As can be seen in Table 3 and Figure 6, the results of the numerical model presented here agrees well with the results obtained by Lelieveld et al. [20] for small time steps up to t=1.5h

In order to compare the results after t=6h and t=12h in Table 3, extrapolation is necessary to reduce calculation time. The approach of extrapolation is briefly explained in the following: The concentration of trace gas (number of virus particles) in each control volume increases from the beginning of the simulation. At a certain point, a quasi-stationary state is established. From this point on, a linear increase of the cumulative particle count function can be assumed for each inhalation volume. This point in time occurs earlier with high ventilation than with low ventilation for the same emission quantity of tracer gas. For high ventilation, the calculation is extended by 10 min to clearly identify that a quasi-stationary state can be assumed after t=90min. Thus, the orange function of the cumulative inhaled particles from Figure 4 and Figure 5 can be continued linearly for each seat. In contrast to the model with high ventilation, the simulation with low ventilation conditions reaches the quasi-stationary state after approx. t=3.75h. The resulting viral load for each seat is then also extrapolated linearly to t=6h and t=12h—see Table 3.

While a good agreement between the results presented and the calculations of Lelieveld et al. [20] can still be seen for short durations of stay, this changes for longer durations. Thus, compared to the presented results of the numerical model, the risk calculator estimates a clearly lower risk of infection. One reason for this overestimation may be that for these large time spans, additional terms must be introduced to deactivate the particles in the numerical model since the airborne survival time of the virus has been exceeded [15]. In the model presented so far, such a particle deactivation is not included. This feature will be considered in future versions of the model.

The calculations of [23] show that after t=6h, 11.5 non-vaccinated people in the room can be infected by just one infectious person (no mask, normal occupancy) in a room of a secondary school. As shown in Table 3, for this calculation, a ventilation rate of $25\;m^{3}/\left(h\cdot \mathrm{person}\right)$ and activities like breathing and talking are assumed. For 16 people in the room, as in the presented study, this would result in an air change rate of $1.667\;h^{-1}$. Considering the case of high ventilation ($432\;m^{3}/h$, $1.800\;h^{-1}$) in the numerical model presented, a ventilation rate of $27\;m^{3}/\left(h\cdot \mathrm{person}\right)$ could be calculated. For an emitted viral load of 100parts/s, 11 susceptible people are exposed to a high risk of infection after t=6h.

Thus, the results of Kriegel and Hartmann [23] and the numerical model after t=6h agree very well. However, the boundary conditions, such as the exact number of susceptible persons in Kriegel and Hartmann [23] and the room size in which they are situated, are not known.

In the case study of Lam-Hine et al. [55], an unvaccinated teacher tests positive for COVID-19 (delta type). Before taking a COVID-19 test, the teacher continued to teach a class over 2 days for about t=6h each. At this time, the teacher could also notice the onset of symptoms of COVID-19 disease. After the test result, 12 of 22 tested unvaccinated students were found to have positive test results within the following 3 days. During class, the students wore a mask but the teacher was temporarily not wearing a mask, reading aloud. The room was ventilated by opened windows on opposite sides. Desks were located throughout the room and at least more than 1.5m apart. Focusing on spatially resolved infection, 8 out of 10 students in the first two rows near the teacher and 4 out of 14 non-vaccinated students sitting in the back three rows became infected.

After t=12h at the latest, in the simulation model, all 15 susceptible persons have a high risk of infection with respect to all analysed cases except for high ventilation and a virus emission of 100parts/s of the infectious person. In Lam-Hine et al. [55], some parameters, such as the room size, ventilation rate, and emitted viral load, are also unknown. It cannot be reliably assumed that the virus is spread exclusively by airborne transmission as the period considered extends over 2 days, and a minimum distance outside the classroom cannot be supposed. Furthermore, an infection of a student beforehand cannot be ruled out. For reasons of comparability of the rooms, the ratio of 12 infected persons out of 22 tested is calculated as 15 persons being susceptible. According to the ratio, a total of 8.2 persons are infected (see Figure 6). Besides that, similar relative results are obtained for the most affected spatial area inside the room; attention should be paid, especially to the quantitative results. In the numerical calculation, all persons in the room exceeded the D50 value. This means that more than half of the persons (7.5) are suffering from the disease. In contrast, there are 8.2 diseases from the outbreak presented, which is in very good agreement with each other. Anyway, the following points should be briefly addressed critically: The distances between people in the models are approximately the same. However, in the model of Lam-Hine et al. [55], there are more people in the classroom, so the volume of the room (and the number of people in it) would have to be increased for a detailed numerical study. The impact of increasing the room volume while decreasing the critical viral dose (due to the delta variant) on the number of individuals at high risk of infection who are additionally exposed to fewer contaminated SARS-CoV-2 particles due to masks should be the subject of further detailed application of the presented model.

In the present study, the same breathing volume and respiration rate were assumed for the students and the teacher. When studying a classroom, especially a younger school class, it must be taken into account that the respiratory volume of children or younger students is less than that of adults [56]. Therefore, a child would inhale fewer virus particles than an adult in a direct comparison, as there are fewer virus particles in a smaller inhalation volume.

The different results in the studies mentioned [20,23] could partly be approximated by the numerical model for selected time periods. If the results on the high risk of infection are interpreted as developing infection, the results can be classified as conservative in comparison. Anyway, independent of the emitted number of particles, a good agreement between the presented numerical model and the results of Lam-Hine et al. [55] can be especially seen concerning the seat-related risk of infection, i.e., which seats are particularly affected over time (see Figure 7).

Anyway, no model is yet known that is validated with experimental results and, thus, could adequately represent the risk of infection. The number of influencing parameters (e.g., activity of the infectious person, number of infectious persons, ventilation strategy, built-in air filter systems, type of face covering, emitted viral load, infectious dose, room occupancy) is very high and very diverse in the different studies, which makes comparability difficult. Clearly, there is a need for experimental validation of numerical predictions—under the same conditions.

## 6. Summary and Conclusions

The presented study has shown that numerical flow simulation (CFD) can be used to estimate the temporally and spatially resolved risk of SARS-CoV-2 infection in a room in the case of exclusive airborne transmission. Therefore, the presented work fills a gap between infection risk assessment and pure numerical simulation on the virus spread of SARS-CoV-2. For this purpose, a numerical model was developed which can be used to deterministically calculate the amount of inhaled virus particles at every seat in the room for a defined emitted viral load of a infectious person. By summing up the amount of inhaled particles per time at specified positions, the time can be determined at which the threshold value for a high risk of infection is reached. Thereby, the duration of stay with a low risk of infection at the different positions correlates with the supply air volume: It has been shown that the duration of stay with still low risk of infection generally becomes longer with increasing supply air volume. Nevertheless, it has also been shown that a rapid distribution of viral particles at high ventilation could be problematic if the viral emission is very high as the infectious dose is quickly reached for a large number of people.

In addition to the calculated infection risk under the current ventilation concept in the room, the numerical flow simulation can be used to investigate possible adjustments of the ventilation to reduce the infection risk spatially and temporally. Basically, the simulation results also show that with increasing distance from the source of infection, the risk of infection decreases at all times in most cases. It could be even determined in comparison to experimental results of other authors, that the results obtained for the evolution of particularly critical locations are in good agreement.

It was found during comparison with other studies that (partly assumed) initial situation and boundary conditions in classrooms could be very different. The necessary adjustments, e.g., due to a required change of the viral emission, are a great advantage of CFD simulations, such as that presented here, since simulations based on new findings of the scientific community can be carried out very quickly. For future research, it would be particularly advisable to obtain reliable data on the infectious dose, so that the threshold of high risk of infection can be adjusted. In addition, a function should be implemented that captures the survival time of SARS-CoV-2 particles in the air if the numerical simulation of longer time periods is subject of interest.

In the study presented, all particles in the inhalation volume are considered for summation. For further investigation, a filter effect through the correct wearing of a mask and, thus, a reduction of the inhaled particles is also possible. Therefore, the model presented here is a base model to analyse particular virus spread in indoor rooms dependent on location and time. It can be used for different applications and virus variants regarding the infection risk potential.

## Figures and Tables

**Figure 1 ijerph-19-06279-f001:**
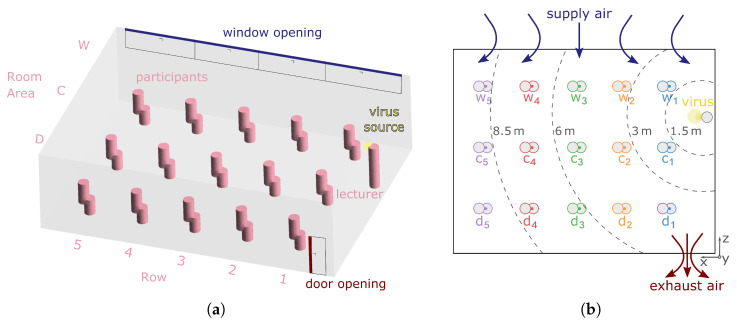
Illustration of the classroom in modelling high cross ventilation of the occupied area: (**a**) side view from the simulation model and (**b**) schematic top view.

**Figure 2 ijerph-19-06279-f002:**
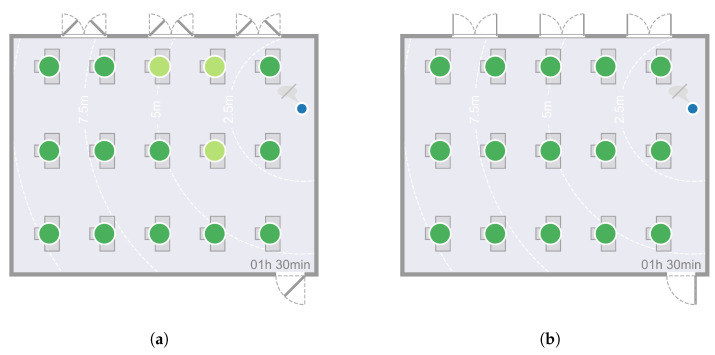
Resulting virus load at each seat after t=90min at low and high ventilation while the infectious lecturer is breathing or speaking. (**a**) Low ventilation, infectious lecturer is breathing (viral load of 10 parts/s). (**b**) High ventilation, infectious lecturer is breathing (viral load of 10 parts/s). (**c**) Low ventilation, infectious lecturer is speaking (viral load of 100 parts/s). (**d**) High ventilation, infectious lecturer is speaking (viral load of 100 parts/s).

**Figure 3 ijerph-19-06279-f003:**
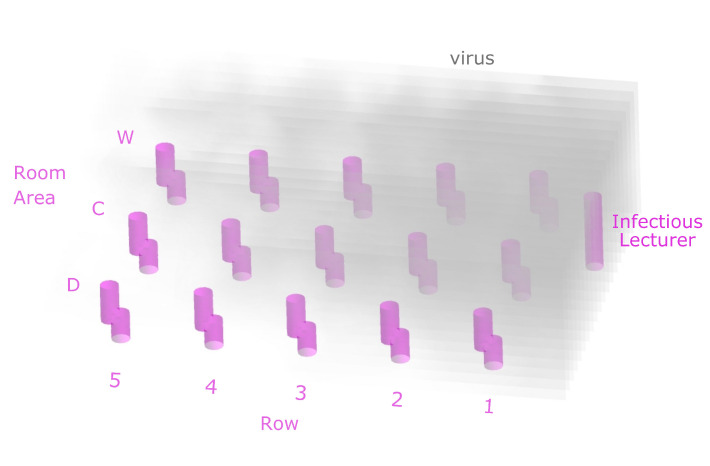
Simulation result of the virus spread in the classroom at low ventilation.

**Figure 4 ijerph-19-06279-f004:**
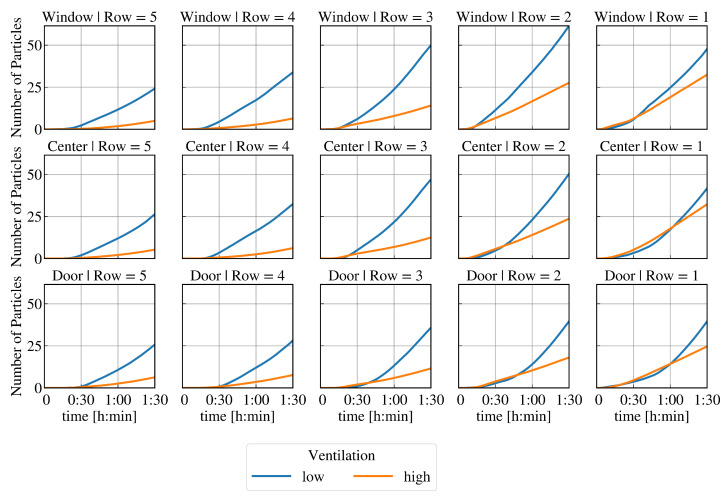
Total inhaled particle count at each seat over time with low and high ventilation and a breathing infectious lecturer (10 parts/s) for the positions in the room (regions at window side, centre, door side; seat rows of 1 to 5—from front to back).

**Figure 5 ijerph-19-06279-f005:**
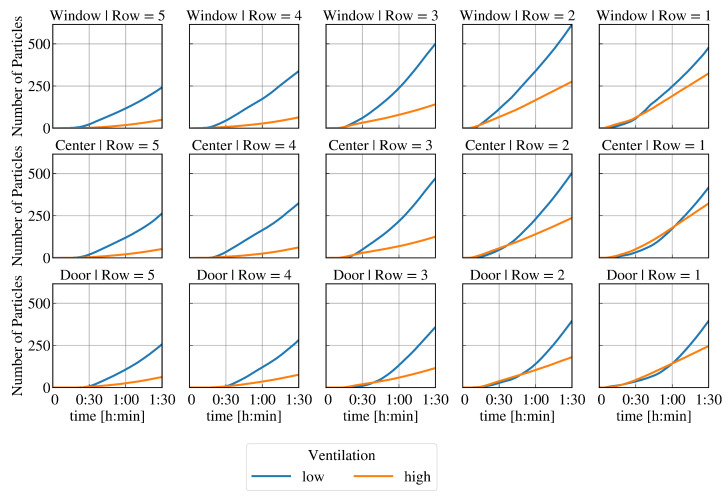
Total inhaled particle count at each seat over time at low and high ventilation and speaking infectious lecturer (100 parts/s) for the positions in the room (regions at window side, center, door side; seat rows of 1 to 5—from front to back).

**Figure 6 ijerph-19-06279-f006:**
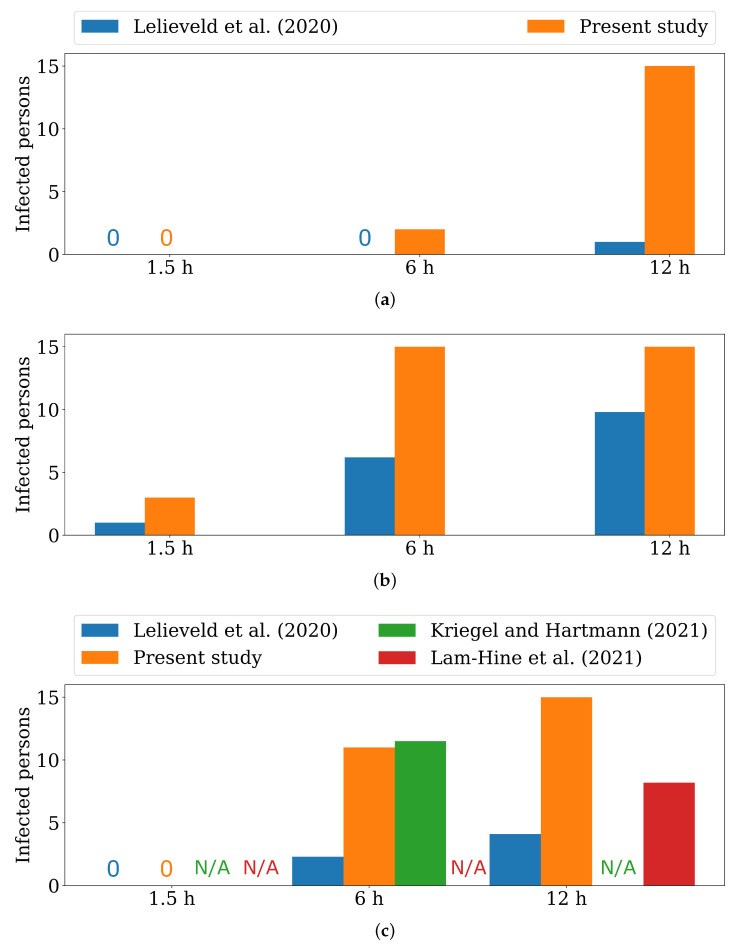
Number of infected persons from available studies compared to the presented study under the conditions from Table 3. (**a**) Results with low ventilation, infectious person breathing [20]. (**b**) Results with low ventilation, infectious person speaking. (**c**) Results with high ventilation, infectious person speaking [20,23,55].

**Figure 7 ijerph-19-06279-f007:**
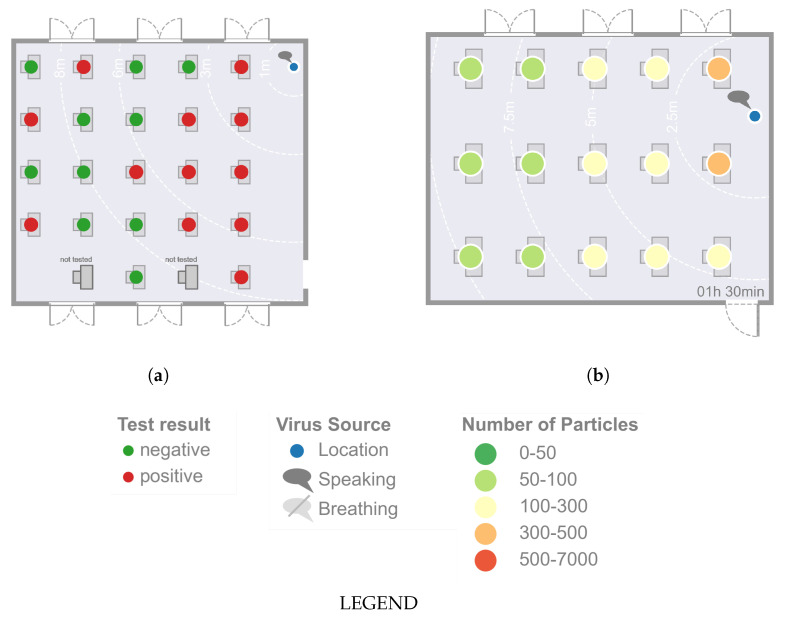
Comparison of the presented numerical model and the results of Lam-Hine et al. [55] with regard to the seat-related risk of infection, i.e., which seats are particularly affected. (**a**) Illustration of infected persons after t=12h in Lam-Hine et al. [55]. (**b**) Resulting virus load at each seat after *t* = 90 min at high ventilation while the infectious lecturer is speaking.

**Table 1 ijerph-19-06279-t001:** Investigated ventilation (vent.) conditions (low/high) and viral loads (speaking/breathing) for the analysed 80 square meter (240 $m^{3}$) classroom with 16 persons inside.

Parameter	Low Vent./ Speaking	High Vent./ Speaking	Low Vent./ Breathing	High Vent./ Breathing
Supply air volume flow in $m^{3}/h$	35	432	35	432
Air change rate in $h^{-1}$	0.146	1.800	0.146	1.800
Viral load of lecturer in virus particles/s	100	100	10	10

**Table 2 ijerph-19-06279-t002:** Model parameters.

Parameter	Description	Value
Model approach		
Virus variant	Virus variant considered in terms of spread and infection	Wildtype
Viral load	Viral emissions of an infectious person during breathing (a) and speaking (b)	(a) 10 SARS-CoV-2 virus particles/s and (b) 100 SARS-CoV-2 virus particles/s
Respiration rate	Respiration rate of all persons	$0.36\;m^{3}/h$
Calculation of viral distribution	Multispecies model approach for calculating the airborne viral load fraction	Species Transport model
Threshold for high risk	The risk of infection is assessed as high as soon as the cumulative number of virus particles in a person’s inhalation volume reaches the threshold value	500 particles
Application model		
Room size	Dimensions of the investigated room	$80\;m^{2}$ (10×8m) and 3m high
Window area	Supply airflow into the room through the surface	$1\;m^{2}$
Supply air	Volume flow of supply air for (a) low ventilation and (b) high ventilation	(a) $35\;m^{3}/h$ and (b) $432\;m^{3}/h$
Open door area	Indoor air escapes through the surface	$0.15\;m^{2}$
Susceptible persons	Number of uninfected persons in the room	15
Distance	Minimum distance between persons	1.5m
Infectious person	Number and location of the infectious persons	1 lecturer, standing at the front of the room
Thermal boundary conditions	Thermal boundary conditions of persons (a), walls (b), fresh air (c), exhaled air (d)	(a) 37 °C, (b) adiabatic, (c) 25 °C, (d) 37 °C
Viscous model	Model for calculating the flow pattern	*k*-ϵ (RNG) model
Numerical mesh	Number of elements of the mesh	Approx. 2.6 million
Near-wall treatment	Near-wall treatment of the room walls that are parallel to the flow direction (from the window to the door) and for those of the bodies of people in the room	Enhanced Wall Treatment with the first near-wall node is set to $y^{+}\approx 1$

**Table 3 ijerph-19-06279-t003:** Important boundary conditions for the risk of infection in the mentioned studies compared to the study presented here with adapted viral load for reasons of comparability.

Parameter	Lelieveld et al. (2020) [20]/MPIC				Kriegel and Hartmann (2021) [23]	Lam-Hine et al. (2021) [55]	Present Study			
**Classroom size in $m^{2}$ **	80				N/A	N/A	80			
**Infectious person**	1 (t=4h before others in room)				1	1 (lecturer)	1 (lecturer)			
**Others**	15 unmasked				Normal occupancy, unmasked	24 masked	15 unmasked			
**Ventilation**	$0.146\;h^{-1}$		$1.800\;h^{-1}$		$25\;m^{3}/\left(h\cdot \mathrm{person}\right)\left(1.667\;h^{-1}\right)$	Windows and door are open; air filter (no ventilation rate given)	$0.146\;h^{-1}$		$1.800\;h^{-1}$	
**Virus variant**	Wildtype				Wildtype	Delta	Wildtype			
**Viral load in parts/s**	10	100	10	100	100	N/A	10	100	10	100
**Result unit**	Infected persons and probability that a least one susceptible person becoming infected				Number of susceptible persons becoming infected	Number of susceptible persons becoming infected	Number of susceptible persons with a high risk of infection			
**Result after 1.5 h**	1.0 (9.4%)	1.0 (86%)	1.0 (3.0%)	1.0 (46%)	No result	No result	0	3	0	0
**Result after 6 h**	1.0 (33%)	6.2	1.0 (11%)	2.2	11.5	No result	2	All 15	0	11
**Result after 12 h**	1.0 (55%)	9.8	1.0 (22%)	4.2	No result	12 (out of 22 tested)	All 15	All 15	0	All 15

## Data Availability

The datasets generated during and/or analysed during the current study are available from the corresponding author upon reasonable request.

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
