# Peer review of "Numerical Flow Simulation on the Virus Spread of SARS-CoV-2 Due to Airborne Transmission in a Classroom"

_ijerph, 2022, doi:10.3390/ijerph19106279_

Round 1

Reviewer 1 Report

After reading the new version of the manuscript “Numerical Flow Simulation on the Virus Spread of SARS-CoV-2 due to Airbone Transmission in a Classroom”, I highlight next remarks:

  • A tiered methodology is necessary to enable replication by other scholars. Current methodological information is dispersed, it should be compiled in a clear and concise manner.
  • Contributions in the body of knowledge and theoretical / practical implications must be outlined in either Introduction or last section.
  • Miscellaneous comments. With the aim of preserving the editorial line of the journal, additional contributions published in the International Journal of Environmental Research and Public Health should be also included. References should not be underlined. Remove the word “see” before citing and please, cite properly according to the guidelines of the journal.

Author Response

Dear Hazel,
dear reviewers,

we would like to thank the reviewers again for the time and effort that they have put into reviewing our revised manuscript. We carefully considered the comments and made additional corrections. As last time, revised text is marked in red in the manuscript. Attached to this letter is our point-to-point response to the reviewers’ comments.

We would like to thank the reviewers and the editor again for taking the time with our manuscript. We are looking forward to your response.

Yours sincerely,
Lara Moeller &
Stephan Schoenfelder

---

Comment:
A tiered methodology is necessary to enable replication by other scholars. Current methodological information is dispersed, it should be compiled in a clear and concise manner.
Answer:
Thanks for the reviewer’s comment. We share the reviewer’s opinion that a tiered description of methodology is very important in order to be able to correctly classify the individual pieces of information in their respective contexts. Therefore, we have differentiated between description of the model (model approach) and application of the model to a specific scenario. To further facilitate reproducibility, Table 2 (page 7) in the updated version of the manuscript is aimed to give a condensed overview of all the essential parameters and settings of the model.

Comment:
Contributions in the body of knowledge and theoretical / practical implications must be outlined in either Introduction or last section.
Answer:
Thanks for the reviewer’s comment. We have decided to move the contribution to the state of the art from the Introduction to the last section (compare lines 69-71 and 465-467). As the theoretical and practical implications are already included there, we completely removed the information given in line 71-74 from the manuscript. Nevertheless, to build a bridge to the following content, we have formulated a slightly more unspecific statement in lines 74-76.

Comment:
Miscellaneous comments. With the aim of preserving the editorial line of the journal, additional contributions published in the International Journal of Environmental Research and Public Health should be also included. References should not be underlined. Remove the word “see” before citing and please, cite properly according to the guidelines of the journal.
Answer:
Thanks for the reviewer’s comments and suggestions. We have included three more articles from the International Journal of Environmental Research and Public Health that address the same topic:
•    Mohamadi, M.; Babington-Ashaye, A.; Lefort, A.; Flahault, A. Risks of Infection with SARS-CoV-2 Due to Contaminated Surfaces: A Scoping Review. Int. J. Environ. Res. Public Health 2021, 18, 11019. doi: 10.3390/ijerph182111019 
– line 27
•    Delikhoon, M.; Guzman, M.I.; Nabizadeh, R.; Norouzian Baghani, A. Modes of Transmission of Severe Acute Respiratory Syndrome-Coronavirus-2 (SARS-CoV-2) and Factors Influencing on the Airborne Transmission: A Review. Int. J. Environ. Res. Public Health 2021, 18, 395. doi: 10.3390/ijerph18020395 –line 28
•    Duill, F.F.; Schulz, F.; Jain, A.; Krieger, L.; van Wachem, B.; Beyrau, F. The Impact of Large Mobile Air Purifiers on Aerosol Concentration in Classrooms and the Reduction of Airborne Transmission of SARS-CoV-2. Int. J. Environ. Res. Public Health 2021, 18, 11523. doi: 10.3390/ijerph182111523 
– line 63

Furthermore, we have removed the word “see” before citing. However, we have not been able to adjust the citation style. We use the official and current LaTeX layout template of the publisher MDPI, which can be found here https://www.mdpi.com/authors/latex (last updated 26 April 2022). Applying the settings necessary for IJERPH (\documentclass[journal,article,submit,pdftex,moreauthors,ijerph]{Definitions/mdpi}) results in this layout. 

Please further note: We have improved the statement in lines 66-69 for clarity.

Reviewer 2 Report

The article gained thanks to the introduced changes significantly. The most valuable changes include:

- a much more detailed description of the assumptions made

- a significant extension of the list of cited publications and justification of the data used in the analyzes

-  taking into account the different infectivity of SARS-CoV-2 variants

- inclusion of the critical elements in the discussion of the obtained results

After the changes, the text can be published in the current form. 

Author Response

Dear Hazel,
dear reviewers,

we would like to thank the reviewers again for the time and effort that they have put into reviewing our revised manuscript. We carefully considered the comments and made additional corrections. As last time, revised text is marked in red in the manuscript. Attached to this letter is our point-to-point response to the reviewers’ comments.

We would like to thank the reviewers and the editor again for taking the time with our manuscript. We are looking forward to your response.

Yours sincerely,
Lara Moeller &
Stephan Schoenfelder

---

Thanks for the reviewer's suggestion for acceptance and the reviewers's contributions to the improvement of our manuscript during the review process.

This manuscript is a resubmission of an earlier submission. The following is a list of the peer review reports and author responses from that submission.

Round 1

Reviewer 1 Report

In this work, a numerical CFD model was developed to calculate the amount of inhaled virus particles regarding the infection risk.

Unfortunately, there exists a large number of similar works which lead and bound this work, and there is simply no novel methodological nor application contributions to be found in this work. Please seriously improve the work before resubmitting again to a reputable peer-reviewed journal.

Author Response

Dear Hazel,
dear reviewers,

thank you for your letter and for the reviewers’ comments concerning our manuscript entitled “Numerical Flow Simulation on the Virus Spread of SARS-CoV-2 due to Airborne Transmission in a Classroom” (Manuscript ID ijerph-1642498). We thank the reviewers for the time and effort that they have put into reviewing the manuscript. Those comments are all valuable and very helpful for revising and improving our paper, as well as the important guiding significance to our researches.
We have carefully studied your comments and have made corrections which we hope will find
approval. Revised text is marked in red in the manuscript. Appended to this letter is our point-to-point response to the reviewers’ comments.

We would like to thank the reviewers and the editor again for taking the time to review our manuscript. We are looking forward to your response.

Yours sincerely,
Lara Moeller &
Stephan Schoenfelder

---

Comment:
In this work, a numerical CFD model was developed to calculate the amount of inhaled virus particles regarding the infection risk.
Unfortunately, there exists a large number of similar works which lead and bound this work, and there is simply no novel methodological nor application contributions to be found in this work. Please seriously improve the work before resubmitting again to a reputable peer-reviewed journal.
Answer:
We thank the reviewer for taking the time to review the manuscript. Regarding your comment, we need to emphasise that, although we are not using a novel method (CFD, Computational Fluid Dynamics), we are using a novel model, as well as, addressing an essential field of its application that has been largely unconsidered so far. Even though scientific contributions concerning the airborne transmission of SARS-CoV-2 based on Computational Fluid Dynamics method in classrooms exist, see, e.g., Mirzaie et al. (2021), Abuhegazy et al. (2020), Löhner et al. (2021) and Ahmadzadeh et al. (2021), however, there is actually no work that aims to provide a temporally and spatially-resolved risk assessment with explicit relation to the infectivity of SARS-CoV-2 in classroom environments. The only known work by Shao et al. (2021) gives an overview of the potential risk of infection in different scenarios (elevator, supermarket, classroom), but does not use SARS-CoV-2 specific data for infection risk assessment, compare Wan et al. (2009) and Morawska et al. (2013). Therefore, the presented work fills a gap between probabilistic infection risk assessment and deterministic numerical simulation on the virus spread of SARS-CoV-2.
We have incorporated this paragraph in a similar form in the manuscript due to the reviewers’ comment (see section Introduction, lines 60-71).

Reviewer 2 Report

After reading the manuscript “Numerical Flow Simulation on the Virus Spread of SARS-CoV-2 due to Airbone Transmission in a Classroom”, I highlight next remarks:

  • Beyond the scope described of the article, the research aim was not defined in the Abstract.
  • Background presented in the first section omits relevant studies linked to the spread of diseases in closed spaces which should serve to identify gaps in the field to be covered by the research, some references are summarized below. With this purpose, an in-depth literature review is required to increase the quantity (21) and quality of references considered by the authors. Research aim should be clearly defined as well.

Sarti D., Campanelli T., Rondina T., Gasperini B. COVID-19 in Workplaces: Secondary Transmission (2021) Annals of Work Exposures and Health, 65 (9), pp. 1145 – 1151.

Birgand G., Peiffer-Smadja N., Fournier S., Kerneis S., Lescure F.-X., Lucet J.-C. Assessment of Air Contamination by SARS-CoV-2 in Hospital Settings (2020) JAMA Network Open, 3 (12), art. no. e2033232

Seto W.H. Airborne transmission and precautions: Facts and myths (2015) Journal of Hospital Infection, 89 (4), pp. 225 – 228

Morawska L., Tang J.W., Bahnfleth W., Bluyssen P.M., Boerstra A., Buonanno G., Cao J., Dancer S., Floto A., Franchimon F., Haworth C., Hogeling J., Isaxon C., Jimenez J.L., Kurnitski J., Li Y., Loomans M., Marks G., Marr L.C., Mazzarella L., Melikov A.K., Miller S., Milton D.K., Nazaroff W., Nielsen P.V., Noakes C., Peccia J., Querol X., Sekhar C., How can airborne transmission of COVID-19 indoors be minimised? (2020) Environment International, 142, art. no. 105832

Liu Y., Ning Z., Chen Y., Guo M., Liu Y., Gali N.K., Sun L., Duan Y., Cai J., Westerdahl D., Liu X., Xu K., Ho K.-F., Kan H., Fu Q., Lan K. Aerodynamic analysis of SARS-CoV-2 in two Wuhan hospitals (2020) Nature, 582 (7813), pp. 557 - 560

  • Because values of SARS-CoV-2 emissions shown in lines 74-79 are very uneven, rationale assumed next by the authors seems debatable. Please provide more relevant arguments to support those assumptions. Grounds to set D50 value and the threshold of high risk for infection also seem unfounded.
  • NO methodology was proposed, instead the mere application of a given model in Section 2 was deemed.
  • The variation of classroom dimensions, the number of students or their position can help to strengthen the study. Please define the notion of “y+ value” (line 150). Arguments to establish values of Table 1 are unknown. Furthermore, more details and their correlation with Section 3 is required for RANS-based k-emodel and k-e (RNG) model.
  • Section 5 should discuss findings obtained in the analysis in comparison to other homogeneous studies in the field. However, Table 2 displays different values of parameters (classroom size, students “others”, ventilation, etc.) which hinder a uniform comparison between them. No further results should be examined herein, i.e., Figure 5 or surplus information about other similar studies such as
  • No conclusions and future lines of research were found in the last section, only a summary of the study. Contributions in the body of knowledge and theoretical / practical implications are unknown as well.
  • Miscellaneous comments. With the aim of preserving the editorial line of the journal, additional contributions published in the International Journal of Environmental Research and Public Health should be also included. Check and complete, if necessary, references [1], [19] and [20]. Tables must appear after being cited, i.e., Table 1. English style should be improved in some parts of the manuscript.

Author Response

Dear Hazel,
dear reviewers,

thank you for your letter and for the reviewers’ comments concerning our manuscript entitled “Numerical Flow Simulation on the Virus Spread of SARS-CoV-2 due to Airborne Transmission in a Classroom” (Manuscript ID ijerph-1642498). We thank the reviewers for the time and effort that they have put into reviewing the manuscript. Those comments are all valuable and very helpful for revising and improving our paper, as well as the important guiding significance to our researches.
We have carefully studied your comments and have made corrections which we hope will find
approval. Revised text is marked in red in the manuscript. Appended to this letter is our point-to-point response to the reviewers’ comments.

We would like to thank the reviewers and the editor again for taking the time to review our manuscript. We are looking forward to your response.

Yours sincerely,
Lara Moeller &
Stephan Schoenfelder

---

After reading the manuscript “Numerical Flow Simulation on the Virus Spread of SARS-CoV-2 due to Airbone Transmission in a Classroom”, I highlight next remarks:
Comment:
Beyond the scope described of the article, the research aim was not defined in the Abstract.
Answer:
We thank the reviewer for taking the time to review the manuscript and the advice. We have refined our Abstract, see lines 5 and 6.

Comment:
Background presented in the first section omits relevant studies linked to the spread of diseases in closed spaces which should serve to identify gaps in the field to be covered by the research, some references are summarized below. With this purpose, an in-depth literature review is required to increase the quantity (21) and quality of references considered by the authors. Research aim should be clearly defined as well.
Answer:
Thanks for pointing out further sources. The work of Liu et al. (2020) was included as a reference in the presented study in line 37. Likewise, the study of Morawska et al. (2020) was referenced in line 37, of Birgand et al. (2020) in line 27, of Seto (2015) in line 26  and of  Sarti et al. (2021) in line 35.  

Comment:
Because values of SARS-CoV-2 emissions shown in lines 74-79 are very uneven, rationale assumed next by the authors seems debatable. Please provide more relevant arguments to support those assumptions. Grounds to set D50 value and the threshold of high risk for infection also seem unfounded.
Answer:
Thanks for the reviewer’s comment. We are sorry that our assumptions have not been made clear enough here. We have added several sources and explanations to the relevant paragraphs (see lines 99-122 for viral emission and lines 165-175 for D50 value).

Comment:
NO methodology was proposed, instead the mere application of a given model in Section 2 was deemed.
Answer:
Thanks for the reviewer’s careful reading and the comment on our methodological approach. Unfortunately, our explanation of the CFD method used was a bit too limited, so we can understand the irritation here. We have now described the method used, within which an ordinary flow model is used, in more detail (see lines 80-89, 201-204 as well as lines 217-224).

Comment:
The variation of classroom dimensions, the number of students or their position can help to strengthen the study. Please define the notion of “y+ value” (line 150). Arguments to establish values of Table 1 are unknown. Furthermore, more details and their correlation with Section 3 is required for RANS-based k-emodel and k-e (RNG) model.
Answer:
Thanks for reading the description of the method carefully. Our aim was to compare the model in this first version with existing models (see answer to the next comment). In order to keep it clear and straight in documentation, we have omitted variations in room size, positions in the room and number of students. In the method presented, only airborne virus transmission via airborne aerosols is used, so that the application of the method is particularly suitable for rooms where people stay at one place for a certain period of time. In the present study, an existing room with its size and occupancy under pandemic conditions was chosen. However, the model and method presented can be used for variations of the variables just mentioned for specific simulations of indoor room situations in future.
In addition, thanks for drawing our attention to the selected boundary conditions of the ventilation. We have found that a transposition has occurred in the ventilation volume. With a high ventilation volume in the presented study of 432 m3/h, this results in an air exchange rate of 27 m3/h*person, which is recommended for schools in Germany according to literature (see line 211, Moriske et al., 2008). The lower ventilation of 35 m3/h is then to be significantly below the high ventilation in order to clearly show the influences with different ventilation rates when applying the method. The low air exchange rate is also below the recommended minimum air exchange rates for buildings. In further investigations, gradations of the ventilation rates or air changes due to pressure differences can be taken into account.
The necessity of using the turbulence model is explained in more detail starting from line 217 ff and the y+ value is also explained more in detail from line 201 onwards.

Comment:
Section 5 should discuss findings obtained in the analysis in comparison to other homogeneous studies in the field. However, Table 2 displays different values of parameters (classroom size, students “others”, ventilation, etc.) which hinder a uniform comparison between them. No further results should be examined herein, i.e., Figure 5 or surplus information about other similar studies such as
Answer:
Thanks for the reviewer’s comment and suggestion. We have moved Figure 5 from section 5 (Discussion) to section 4 (Results). We agree with the reviewer that a uniform comparison seems difficult at first. Our model is based on a classroom that we have available at our university for teaching. However, the comparability is as good as the data available from the other models. We would like to note the following in detail:
•    Classroom size: Our room size of 80 square meters can also be resolved by the model of Lelieveld et al. (2020). For the other two models (Kriegel and Hartmann (2021), Lam-Hine et al. (2021)) the room size is unknown. 
•    Infectious person: All models or the documented outbreak include one infectious person.
•    Others: In the model of Lelieveld et al. (2020) and its risk calculator, a specific room occupancy can be defined; for example, a room occupancy of 15 persons can also be modeled. Kriegel and Hartmann (2021) do not quantify room occupancy, but only specify a qualitatively "normal occupancy."  We agree with the reviewer's critique on this point for the comparison with the outbreak described by Lam-Hine et al. (2021). There were 24 persons in the room. The same applies to the viral variant that we newly report in our manuscript. All models refer to the wild type except the outbreak documented by Lam-Hine et al. (2021). We have critically addressed this point in Manuscript Lines 434-441.
•    Ventilation: Ventilation conditions are identical (compare Lelieveld et al., 2021) or very similar (compare Kriegel and Hartmann (2021)). Only Lam-Hine et al. (2021) do not specify ventilation conditions. The situation is similar for the viral load.  
•    Result unit: Here, Lelieveld et al. (2021) assume the D50 value, as we do. Lelieveld only assume the risk above 50% as probable infection. Hartmann and Kriegel's evaluation is based on the Wells-Riley method, assuming a risk above 63.2% as possible infection. 
Since our model has not yet been experimentally validated, we rely on the comparison in this table. Without those data, validation is missing. We are planning an experimental validation, but cannot offer it at this stage. Though, the model already works for general analysis of possible virus distribution, leading to better understanding of indoor air fluid dynamics and virus distribution. Thus, we think, the results so far are worth to be published as one step for better modelling and understanding spatially virus distribution. 
Comment:
No conclusions and future lines of research were found in the last section, only a summary of the study. Contributions in the body of knowledge and theoretical / practical implications are unknown as well.
Answer:
The last section has been renamed and is now titled Summary and Conclusion. Further examples of possible conditions for future investigations are now included in this section (see lines 489 ff. and line 496). The usefulness of the method, further possible applications of the method for theoretical / practical implications and adaptations of the models in the case of new findings in science have already been explained in the discussion from line 329 ff. onwards and still subsequently included in the abstract in lines 5-6.

Comment:
Miscellaneous comments. With the aim of preserving the editorial line of the journal, additional contributions published in the International Journal of Environmental Research and Public Health should be also included. Check and complete, if necessary, references [1], [19] and [20]. Tables must appear after being cited, i.e., Table 1. English style should be improved in some parts of the manuscript.
Answer:
The references mentioned have been revised and completed (see line 513, 625 and 627). The citations on Table 1 have also been edited.

Reviewer 3 Report

Dear authors!

The simulation model presented in your manuscript is described carefully and discussed in detail. However, extrapolation to 6 and 12 hours of stay in the room raises questions on the assumptions, at least on the condition of the lecturer, since the number of virus particles in the exhaled air of an infected person, and even more so in a sick person, will change significantly during this time.

Author Response

Dear Hazel,
dear reviewers,

thank you for your letter and for the reviewers’ comments concerning our manuscript entitled “Numerical Flow Simulation on the Virus Spread of SARS-CoV-2 due to Airborne Transmission in a Classroom” (Manuscript ID ijerph-1642498). We thank the reviewers for the time and effort that they have put into reviewing the manuscript. Those comments are all valuable and very helpful for revising and improving our paper, as well as the important guiding significance to our researches.
We have carefully studied your comments and have made corrections which we hope will find
approval. Revised text is marked in red in the manuscript. Appended to this letter is our point-to-point response to the reviewers’ comments.

We would like to thank the reviewers and the editor again for taking the time to review our manuscript. We are looking forward to your response.

Yours sincerely,
Lara Moeller &
Stephan Schoenfelder

---

Comment:
Dear authors!
The simulation model presented in your manuscript is described carefully and discussed in detail. However, extrapolation to 6 and 12 hours of stay in the room raises questions on the assumptions, at least on the condition of the lecturer, since the number of virus particles in the exhaled air of an infected person, and even more so in a sick person, will change significantly during this time.
Answer:
Thanks for taking the time to review the manuscript and especially for reading the methods used carefully. This medical and physiological condition was not considered in the study presented since our model was basically intended for normal teaching intervals of around 90 mins (where this fact can be ignored). With the current application of the method for the classroom (concerning long time periods), a very conservative case is examined. This means that the risk of infection in the classroom is examined under the respective conditions in the worst case. For subsequent investigations using the presented method, a consideration of the temporal change of the emersion values is planned. We have now mentioned this in the discussion section for clarification (see line 392 ff.).

Reviewer 4 Report

The topic of the article is very important and up-to-date. The applied approach is interesting and valuable. Unfortunately, there are significant misunderstandings, simplifications, and a lack of consistency in the introduction and description of the model.

Lines 23-24: „It is now considered certain that airborne transmission is the primary transmission mechanism of SARS-CoV-2 [see 26]. Generally, the sentence is correct; however, the literature (some published before the SARS-CoV-2 pandemic) supporting this statement is wrongly selected. The examples of much more relevant references addressing the problem of whether airborne is dominating infection way of transmission:

  • Correia, G., Rodrigues, L., Da Silva, M. G., & Gonçalves, T. (2020). Airborne route and bad use of ventilation systems as non-negligible factors in SARS-CoV-2 transmission. Medical hypotheses, 141, 109781.
  • Stabile, L., Pacitto, A., Mikszewski, A., Morawska, L., & Buonanno, G. (2021). Ventilation procedures to minimize the airborne transmission of viruses in classrooms. Building and environment, 202, 108042.
  • Tang, J. W., Bahnfleth, W. P., Bluyssen, P. M., Buonanno, G., Jimenez, J. L., Kurnitski, J., ... & Dancer, S. J. (2021). Dismantling myths on the airborne transmission of severe acute respiratory syndrome coronavirus-2 (SARS-CoV-2). Journal of Hospital Infection, 110, 89-96.
  • Peng, Z., Rojas, A. P., Kropff, E., Bahnfleth, W., Buonanno, G., Dancer, S. J., ... & Jimenez, J. L. (2022). Practical indicators for risk of airborne transmission in shared indoor environments and their application to covid-19 outbreaks. Environmental science & technology.
  • Mikszewski, A., Stabile, L., Buonanno, G., & Morawska, L. (2022). Increased close proximity airborne transmission of the SARS-CoV-2 Delta variant. Science of The Total Environment, 816, 151499.

The authors complete input data from different studies. Therefore, it is essential to check what they are precisely mean and whether quoted data are reliable.

Lines 74-82: „The measurements of Hartmann et al. [13] show an emission of 134 SARS-CoV-2 virus particles/s when breathing and 195 SARS-CoV-2-virus particles/s when speaking. In Coleman et al. [14] approx. 10.8 SARS-CoV-2-virus particles/s are emitted when breathing and 81.6 SARS-CoV-2-virus particles/s when speaking.1 The calculations in Lelieveld et al. [9] assume a viral load up to 10 SARS-CoV-2-virus particles/s and 100 SARS-CoV-2-virus particles/s for breathing and speaking activities, respectively. Following the studies presented, within the work presented here a viral emission of 10 SARS-CoV-2 virus particles/s during breathing and 100 SARS-CoV-2 virus particles/s for speaking is assumed.”

This paragraph changes the sense of the original study by Hartmann, et al. They define the number of sputum particles, not the number of „SARS-CoV-2 virus particles”. Please note that the viral load concentrations in the mouth can reach 109 RNA copies mL-1(occasionally up to 1011 RNA copies mL−1) during the course of the disease.

Typically infection risk is calculated based on Wells-Riley model. Such an approach used the concepts of quantum (the dose of airborne droplet nuclei required to cause infection in 63% of susceptible persons). Quanta values are different for SARS-CoV- 2 virus variants (Alfa, Beta, Gamma, Delta, or Omicron). For instance, it is assumed that for Delta variant, transmissivity is approximately 2 times higher, while for Omikron approx. 4 times higher than for the original SARS-CoV-2 virus. Similar phenomena should be applied to the risk model used in the reviewed study.

Unfortunately, the authors do not specify what variant they are modeling, which is a significant oversight. Moreover, they do not sufficiently check what variant was investigated in the quoted references.

Some doubts are related to the assumed amount of respiration 0,36 m3/h [15]. It seems to be strongly underestimated. Please compare:

  • Allan, M., & Richardson, G. M. (1998). Probability density functions describing 24-hour inhalation rates for use in human health risk assessments. Human and Ecological Risk Assessment: An International Journal, 4(2), 379-408.

Moreover, it is worth noting that standing teachers and sitting pupils (male and female of different ages) will have different respiration rates.

Considering severe doubts related to the assumption, evaluating the results and conclusions is pointless.

Author Response

Dear Hazel,
dear reviewers,

thank you for your letter and for the reviewers’ comments concerning our manuscript entitled “Numerical Flow Simulation on the Virus Spread of SARS-CoV-2 due to Airborne Transmission in a Classroom” (Manuscript ID ijerph-1642498). We thank the reviewers for the time and effort that they have put into reviewing the manuscript. Those comments are all valuable and very helpful for revising and improving our paper, as well as the important guiding significance to our researches.
We have carefully studied your comments and have made corrections which we hope will find
approval. Revised text is marked in red in the manuscript. Appended to this letter is our point-to-point response to the reviewers’ comments.

We would like to thank the reviewers and the editor again for taking the time to review our manuscript. We are looking forward to your response.

Yours sincerely,
Lara Moeller &
Stephan Schoenfelder

---

The topic of the article is very important and up-to-date. The applied approach is interesting and valuable. Unfortunately, there are significant misunderstandings, simplifications, and a lack of consistency in the introduction and description of the model.
Comment:
Lines 23-24: „It is now considered certain that airborne transmission is the primary transmission mechanism of SARS-CoV-2 [see 2–6]. Generally, the sentence is correct; however, the literature (some published before the SARS-CoV-2 pandemic) supporting this statement is wrongly selected. The examples of much more relevant references addressing the problem of whether airborne is dominating infection way of transmission:
•    Correia, G., Rodrigues, L., Da Silva, M. G., & Gonçalves, T. (2020). Airborne route and bad use of ventilation systems as non-negligible factors in SARS-CoV-2 transmission. Medical hypotheses, 141, 109781.
•    Stabile, L., Pacitto, A., Mikszewski, A., Morawska, L., & Buonanno, G. (2021). Ventilation procedures to minimize the airborne transmission of viruses in classrooms. Building and environment, 202, 108042.
•    Tang, J. W., Bahnfleth, W. P., Bluyssen, P. M., Buonanno, G., Jimenez, J. L., Kurnitski, J., ... & Dancer, S. J. (2021). Dismantling myths on the airborne transmission of severe acute respiratory syndrome coronavirus-2 (SARS-CoV-2). Journal of Hospital Infection, 110, 89-96.
•    Peng, Z., Rojas, A. P., Kropff, E., Bahnfleth, W., Buonanno, G., Dancer, S. J., ... & Jimenez, J. L. (2022). Practical indicators for risk of airborne transmission in shared indoor environments and their application to covid-19 outbreaks. Environmental science & technology.
•    Mikszewski, A., Stabile, L., Buonanno, G., & Morawska, L. (2022). Increased close proximity airborne transmission of the SARS-CoV-2 Delta variant. Science of The Total Environment, 816, 151499.
Answer:
We thank the reviewer for taking the time to review the manuscript and for the valuable references to support airborne virus transmission. We have referenced all the recommended studies in this paper. The publication of Correia et al. (2020) refers to the discussion of airborne virus transmission at the beginning of the SARS-CoV-2 pandemic in line 26. The study by Stabile et al. (2021) and Tang et al. (2021)/Peng et al. (2022) are considered in line 40 and 27, respectively. The reference by Mikszewski (2022) can now be found in line 169.

Comment:
The authors complete input data from different studies. Therefore, it is essential to check what they are precisely mean and whether quoted data are reliable.
Lines 74-82: „The measurements of Hartmann et al. [13] show an emission of 134 SARS-CoV-2 virus particles/s when breathing and 195 SARS-CoV-2-virus particles/s when speaking. In Coleman et al. [14] approx. 10.8 SARS-CoV-2-virus particles/s are emitted when breathing and 81.6 SARS-CoV-2-virus particles/s when speaking.1 The calculations in Lelieveld et al. [9] assume a viral load up to 10 SARS-CoV-2-virus particles/s and 100 SARS-CoV-2-virus particles/s for breathing and speaking activities, respectively. Following the studies presented, within the work presented here a viral emission of 10 SARS-CoV-2 virus particles/s during breathing and 100 SARS-CoV-2 virus particles/s for speaking is assumed.”
This paragraph changes the sense of the original study by Hartmann, et al. They define the number of sputum particles, not the number of „SARS-CoV-2 virus particles”. Please note that the viral load concentrations in the mouth can reach 109 RNA copies mL-1(occasionally up to 1011RNA copies mL−1) during the course of the disease.
Answer:
Thanks to the reviewer for pointing out this error. We are sorry for this mistake. We have corrected the corresponding paragraph, see Lines 99-122.

Comment:
Typically, infection risk is calculated based on Wells-Riley model. Such an approach used the concepts of quantum (the dose of airborne droplet nuclei required to cause infection in 63% of susceptible persons). Quanta values are different for SARS-CoV- 2 virus variants (Alfa, Beta, Gamma, Delta, or Omicron). For instance, it is assumed that for Delta variant, transmissivity is approximately 2 times higher, while for Omikron approx. 4 times higher than for the original SARS-CoV-2 virus. Similar phenomena should be applied to the risk model used in the reviewed study.
Answer:
Thanks for the reviewer’s careful reading for helping us to improve this paper. We have included the aspect of different virus variants, compare lines 165-175. In addition, this aspect is critically addressed due to the comparison with the model of Lam-Hine et al. (2021), see lines 434-447.

Comment:
Unfortunately, the authors do not specify what variant they are modeling, which is a significant oversight. Moreover, they do not sufficiently check what variant was investigated in the quoted references.
Answer:
Thanks for the reviewer’s comment. We have now explained at various points in the text which virus variant is involved in each case (see, e.g., lines 109, 122, 167-173, 411, 439 and Table 2).

Comment:
Some doubts are related to the assumed amount of respiration 0,36 m3/h [15]. It seems to be strongly underestimated. Please compare:
•    Allan, M., & Richardson, G. M. (1998). Probability density functions describing 24-hour inhalation rates for use in human health risk assessments. Human and Ecological Risk Assessment: An International Journal, 4(2), 379-408.
Answer:
Thanks for the reviewer’s suggestion. We would like to note, that the persons modelled are at rest. Therefore, the amount of respiration is orientated rather to the low values within the range of average respiratory volumes, see source Hyldgaard et al. (1994), König et al. (2007), Chourpiliadis et al. (2022); Further reference (see König et al. (2007), Chourpiliadis et al. (2022)) on the magnitude of respiration above 0.36 m3/h was added (compare line 126). In the study of Allen & Richardson (1998) 24-hour inhalation rates of different ages and sexes were studied. However, total respiration over a day of a person and differentiation by sex and age are not part of the presented study but could be considered in further studies, insofar as these aspects are to be investigated separately.

Comment:
Moreover, it is worth noting that standing teachers and sitting pupils (male and female of different ages) will have different respiration rates.
Answer:
Thanks for the reviewer’s comment. In the study presented, the influence of gender on respiratory volume is neglected in order to compare positions in the room in terms of infection risk. All persons in the room have the same respiratory volume. A distinction to the teaching person, as well as any other person in the room, is possible with the presented model and can be used in subsequent investigations. 
However, the respiratory volume of the infectious person as a boundary condition of the simulation is not decisive for the emitted virus particles. The viral load of the infectious person with the presented approach is varied in the evaluation depending on the activity level. Since it is not clear how old the students are, we assumed the same respiratory volume for all individuals for simplicity.  For young children, the amount of virus inhaled would tend to decrease. We have added this aspect in the discussion (see line 442 ff). In this regard, we have added the study of Allen & Richardson (1998) as a reference, as it is appropriate here.

Comment:
Considering severe doubts related to the assumption, evaluating the results and conclusions is pointless.
Answer:
We hope that we could clearify our assumptions now. In general, we want to emphasise that we are presenting a model and method to calculate and visualise virus distribution in indoor rooms considering the deterministic people risk of infection. Of course, the model so far needs to be simplified for general understanding of the first results here. Though, the model and method is able to consider all the different aspects mentioned by the reviewer, which can be used in different future application.